# cGAS-STING dependent type I IFN reduces *Leptospira interrogans* renal colonization in mice

**Suman Gupta**[1,2], **James Matsunaga**[3,4], **Bridget Ratitong**[1,2], **Andrew Manion**[5],
**Sana Ismaeel**[6], **Diogo G. Valadares**[1,2], **A. Phillip West**[7], **Nagaraj Kerur**[8], **Christian Stehlik**[6,9],
**Andrea Dorfleutner**[6,9], **Jargalsaikhan Dagvadorj**[1,2], **Jenifer Coburn**[10,11], **Andrea J. Wolf**[9,12],
**Meghan A. Morrissey**[13], **Suzanne L. Cassel**[1,2,9☯*], **David A. Haake**[4,14☯*],
**Fayyaz S. Sutterwala** [1,2,4,9☯*]

1 Department of Medicine, Cedars-Sinai Medical Center, Los Angeles, California, United States of America, 2 Women's Guild Lung Institute, Cedars-Sinai Medical Center, Los Angeles, United States of America, 3 Research Service, VA Greater Los Angeles Healthcare System, Los Angeles, California, United States of America, 4 Department of Medicine, David Geffen School of Medicine, University of California, Los Angeles, Los Angeles, California, United States of America, 5 Interdisciplinary Program in Quantitative Biosciences, University of California, Santa Barbara, Santa Barbara, California, United States of America, 6 Department of Pathology and Laboratory Medicine, Cedars-Sinai Medical Center, Los Angeles, California, United States of America, 7 The Jackson Laboratory, Bar Harbor, Maine, United States of America, 8 Department of Ophthalmology and Visual Sciences, The Ohio State University Wexner Medical Center, Columbus, Ohio, United States of America, 9 Department of Biomedical Sciences, Cedars-Sinai Medical Center, Los Angeles, California, United States of America, 10 Department of Medicine, Medical College of Wisconsin, Milwaukee, Wisconsin, United States of America, 11 Department of Microbiology and Immunology, Medical College of Wisconsin, Milwaukee, Wisconsin, United States of America, 12 F. Widjaja Inflammatory Bowel Disease Institute, Cedars-Sinai Medical Center, Los Angeles, California, United States of America, 13 Molecular Cellular and Developmental Biology Department, University of California, Santa Barbara, California, United States of America, 14 Division of Infectious Diseases, VA Greater Los Angeles Healthcare System, Los Angeles, California, United States of America

☯ These authors contributed equally
* suzanne.cassel@cshs.org (SLC), DHaake@UCLA.edu (DAH), fayyaz.sutterwala@cshs.org (FSS)

## Abstract

*Leptospira interrogans* is the major causative agent of leptospirosis. Humans, canines and agricultural animals are susceptible to *Leptospira* species and can develop fulminant disease. Rodents serve as reservoir hosts in which the bacteria colonize the renal tubules and are excreted in the urine. The host immune response to *Leptospira* spp. remains poorly defined. We show that *L. interrogans* induces a robust type I interferon (IFN) response in human and murine macrophages that is dependent on the cytosolic dsDNA sensor Cyclic GMP-AMP Synthase (cGAS) and the Stimulator of IFN Genes (STING) signaling pathway. Further, we show that mice deficient in the IFNα/β receptor subunit 1 (IFNAR1) or STING had higher bacterial burdens and increased renal colonization following infection *in vivo* suggesting that cGAS-STING-driven type I IFN is required for the host defense against *L. interrogans*. These findings demonstrate the significance of cGAS-STING- dependent type I IFN signaling in mammalian innate immune responses to *L. interrogans*.

**Data availability statement:** All relevant data generated in this study are present within the manuscript and supplemental information. All data underlying the findings in the study have been made available at: https://zenodo.org/records/17797875 DOI: 10.5281/zenodo.17797875.

**Funding:** This work was supported by National Institutes of Health grants P01 AI168148 (D.A.H.), R01 AI175101 (F.S.S and J.D.), R01 AI177233 (F.S.S and S.L.C.), and R35 GM146935 (M.A.M.). M.A.M. is a Nadia's Gift Foundation Innovator of the Damon Runyon Cancer Research Foundation (DRR-85-25). The funders had no role in study design, data collection and analysis, decision to publish, or preparation of the manuscript.

**Competing interests:** The authors have declared that no competing interests exist.

## Author summary

Leptospirosis is a zoonotic disease primarily caused by infection with the *Leptospira interrogans* bacteria. The manifestation of this infection is variable depending on the type of animal infected. For example, while the disease can be very severe and even fatal in people, livestock, and dogs, rodents do not die or become significantly ill after infection. Instead, mice and rats develop persistent but asymptomatic colonization by the bacteria within their kidneys. They then shed the bacteria in their urine, allowing the infection to spread through the environment to other animals. Despite the significant global burden of leptospirosis, the innate immune pathways that detect this pathogen and regulate renal colonization remain poorly understood. In this study we demonstrate that *L. interrogans* induces a robust innate immune response from macrophages with release of pro-inflammatory type I IFNs after the sensing of cytosolic DNA. Using *in vivo* mouse models of *L. interrogans* infection we further show that activation of this pathway is required to control bacterial burdens and reduce long-term kidney colonization. This study is the first to demonstrate a critical role for cytosolic DNA sensing and type I IFN in controlling *L. interrogans* infection.

## Introduction

*L. interrogans* is responsible for most pathogenic cases of Leptospirosis [1]. All vertebrates are susceptible to infection, and the manifestation of symptoms varies extensively in different hosts [2]. Humans are incidental hosts with many infections remaining asymptomatic. However, humans can develop multiorgan dysfunction which can prove fatal in 10–20% of cases [3]. In contrast, rodents such as mice and rats typically remain asymptomatic while being chronically colonized in their kidneys, serving as reservoir hosts for *Leptospira* spp. and shedding the bacteria in their urine for long periods of time [4–6]. This process leads to soil and water contamination, perpetuating the enzootic cycle of the disease.

The role of macrophages in the detection and clearance of leptospires remains incompletely understood [7]. Depletion of peritoneal macrophages in C57BL/6J mice was shown to increase sensitivity to leptospirosis, resulting in an increased bacterial burden [8–11]. In addition, macrophage depletion led to increased kidney colonization [10,11]. Leptospires are generally classified as extracellular bacteria, although recent studies have suggested the presence of leptospires within macrophages [12–14]. Leptospires may exist freely in the cytosol or reside within specific cellular compartments, such as endosomes, lysosomes, or phagosomes [13]. The fate of internalized leptospires is unclear, but recent findings indicate that leptospires do not replicate within mouse macrophages and that viable internalized leptospires can exit these cells over time [15]. Overall, macrophages likely shape leptospiral infection through mechanisms that are multifaceted and not yet fully defined.

Intracellular innate immune pathways that sense the presence of cytosolic nucleic acids can play a critical role in the generation of type I interferon (IFN) responses. Cyclic GMP-AMP synthase (cGAS) is one such intracellular sensor that can bind to double-stranded DNA in the cytosol [16–20]. Upon binding, cGAS generates 2'3'-cyclic guanosine monophosphate-adenosine monophosphate (cGAMP), which activates stimulator of interferon genes (STING). This activation results in the recruitment and subsequent phosphorylation of Tank-binding kinase-1 (TBK1) and IFN regulatory factor 3 (IRF3), followed by the production and secretion of IFNα and IFNβ. Type I IFN signaling through the IFNα/β receptor (IFNAR) induces the transcription of interferon-stimulated genes (ISGs) [21,22]. The cGAS-STING pathway was initially shown to play a role in anti-viral host response; it has since been shown to have roles in immune responses to a variety of bacterial pathogens including *Listeria monocytogenes*, *Mycobacterium tuberculosis*, *Klebsiella pneumoniae*, and *Borrelia burgdorferi* [17,19,20,23]. Bacteria that produce cyclic dinucleotides can directly activate STING independently from cGAS [24–27]. Whether *Leptospira* can activate the cGAS-STING pathway and the physiologic relevance of type I IFN production in leptospirosis is currently unknown.

Because pathogenic leptospires possess genes involved in cyclic dinucleotide synthesis [28,29] and leptospires can be internalized by macrophages, we asked if cGAS-STING-mediated type I IFN signaling was involved in the host immune responses against leptospiral infection. In this study, we demonstrate that *L. interrogans* induces type I IFN production and the transcription of ISGs in both mouse and human macrophages. The type I IFN response was found to be dependent on effective phagocytosis and degradation of leptospires in the macrophages. Further, we show that the type I IFN induction in *L. interrogans*-infected macrophages involves the cGAS-STING pathway. Finally, the absence of IFNAR or STING *in vivo* resulted in increased susceptibility to *L. interrogans* infection and increased kidney colonization, highlighting the importance of cGAS-STING-driven type I IFN in mammalian host defense against *L. interrogans*.

## Results

### *Leptospira* induces a type I IFN response following infection of macrophages

We first evaluated the ability of mouse macrophages to induce a type I IFN response following infection with *L. interrogans*. Consistent with prior reports we observed that bone marrow-derived macrophages (BMDMs) infected with *L. interrogans* serovar Copenhageni strain Fiocruz L1-130 at a multiplicity of infection (MOI) of 100 elicited the secretion of TNF, IL-10, and IL-6 (**Fig 1A**-**1C**) [30,31]. Additionally, robust secretion of IFNβ was also observed (**Figs 1D and** S1A). Tank-binding kinase 1 (TBK1) and IFN regulatory factor 3 (IRF3) play central roles in the initiation of type I IFN responses following signaling by the mitochondrial antiviral signaling protein (MAVS), STING, and TICAM1 [32]. We found *L. interrogans* infection of BMDM resulted in the phosphorylation of TBK1 and IRF3 consistent with the observed production of IFNβ (**Fig 1E**). We also assessed the transcript levels of *Ifna* and *Ifnb* in BMDMs and observed upregulation of both transcripts by 4 hours post-infection (**Fig 1F**, **1G**). Similar kinetics for IFN-β transcript upregulation were observed in PMA-differentiated human THP-1 cells (**Fig 1H**) suggesting *L. interrogans* can induce a type I IFN response in both mouse and human macrophages.

### Phagocytic uptake of *L. interrogans* is required for effective IFNβ secretion

To determine if *L. interrogans* were being phagocytosed, BMDMs were infected with pHrodo-labelled bacteria and increase in fluorescence dependent on phagosomal acidification was assessed. We found a robust increase in florescence of pHrodo-labelled bacteria with time suggesting bacteria were being taken up in an acidic phagosomal compartment (**Fig 2A**). Additionally, wild-type (WT) BMDMs pretreated with cytochalasin D, an inhibitor of actin polymerization, prior to infection with pHrodo-labelled *L. interrogans* showed a reduction in fluorescence, suggesting that the uptake of bacteria into acidic subcellular compartments is via a phagocytic pathway (**Fig 2A**). After phagocytosis, bacteria-containing phagosomes recruit Rab5 to their surface, which is then replaced by LAMP1 upon lysosomal fusion [33–36].

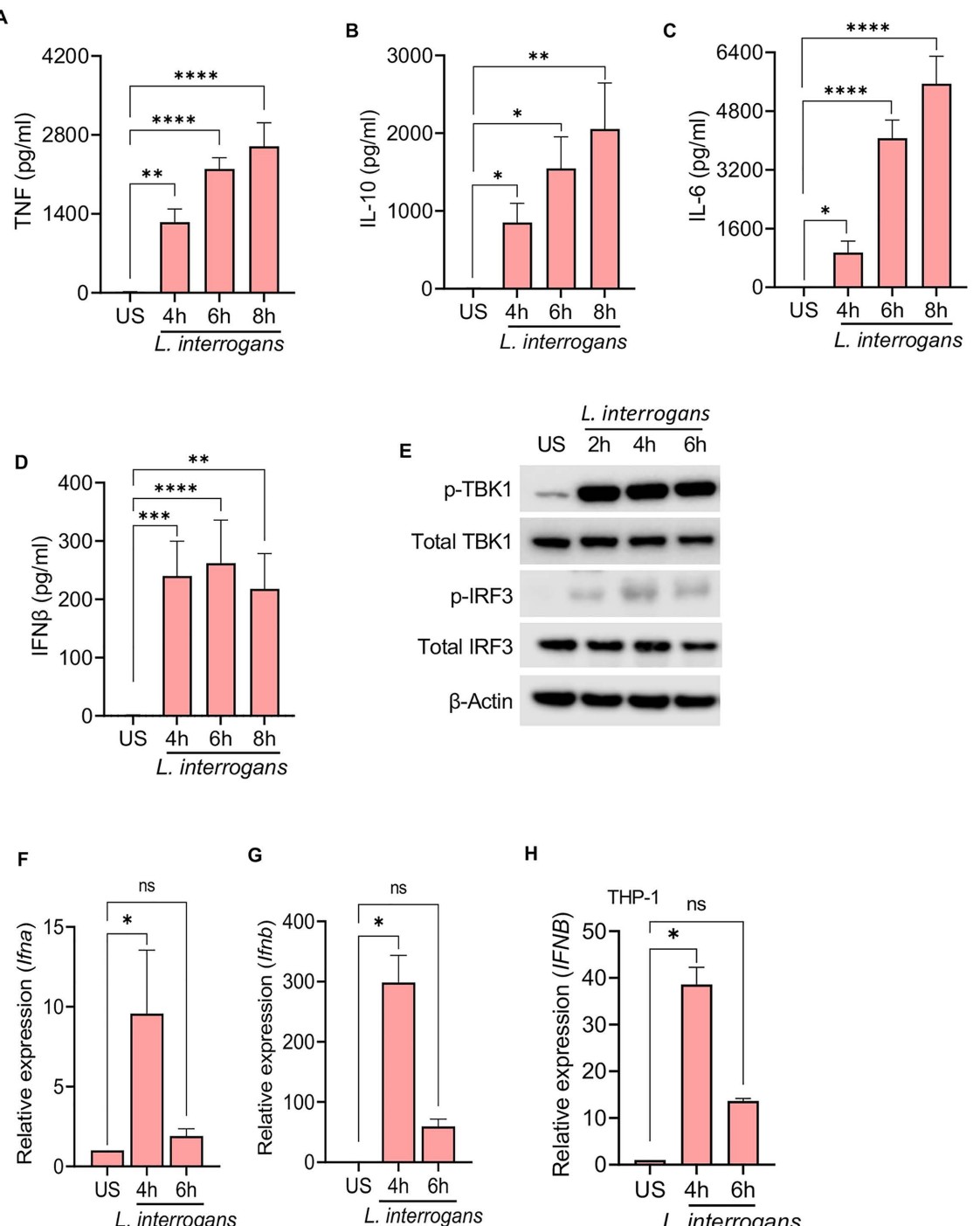

**Fig 1. *L. interrogans* infection of macrophages induces a type I IFN response.** (A-D) Quantification of TNF, IL-10, IL-6, and IFNβ by ELISA in culture supernatants of WT BMDMs infected with *L. interrogans* Fiocruz L1-130 (MOI 100) for 4, 6, and 8h. Data are pooled from three independent experiments with technical triplicates and plotted as the mean ± SEM. (E) Immunoblot for p-TBK1, total TBK1, p-IRF3, total IRF3 and β-actin in cell lysates of BMDMs infected with *L. interrogans* (MOI 100) for 2, 4, and 6h. Data are representative of three independent experiments. (F, G) Relative expression of *Ifna* and *Ifnb* transcript levels assessed by qPCR in BMDMs infected with *L. interrogans* at 4 and 6h post infection. (H) Relative expression of *IFNB* transcript levels assessed by qPCR in PMA-differentiated human THP-1 cells infected with *L. interrogans* at 4, 6, and 8h post infection. (F-H) Data are pooled from three independent experiments with technical triplicates and expressed as the mean ± SEM. Statistical significance calculated by one-way ANOVA; *p<0.05, **p<0.01, ***p<0.001, ****p<0.0001, ns=non-significant. US=unstimulated.

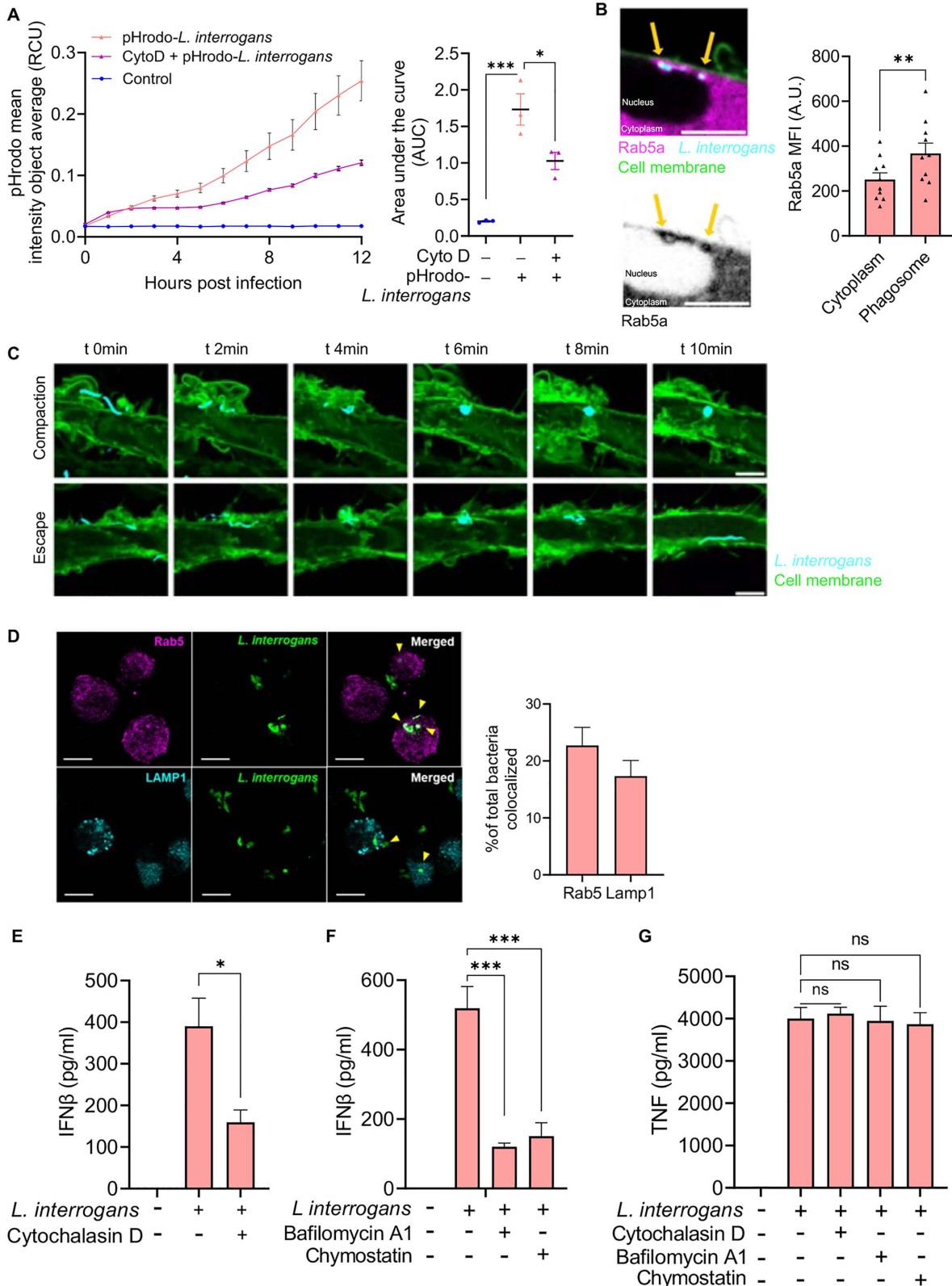

**Fig 2. Phagocytosis of *L. interrogans* is required for macrophage driven IFNβ production.** (A) Increase in mean fluorescence intensity per cell (Red Calibrated Unit; RCU) in WT BMDMs, after addition of *L. interrogans* Fiocruz L1-130 (MOI 100) labelled with pH-sensitive dye (pHrodo Red,

succinimidyl ester), was read in an Incucyte SX5 reader at 1h intervals for a period of 12h in the presence or absence of cytochalasin D (2μM). Curves are representative of three independent experiments and are the mean ±SD of technical triplicates. Area under the curve (AUC) was calculated for the increase in pHrodo fluorescence intensity per cell in WT BMDMs in the presence or absence of cytochalasin D. Data are pooled from three independent experiments with technical triplicates and expressed as the mean±SEM. (B) BMDMs expressing Rab5a-mCherry (magenta) and a membrane tethered GFP (GFP-CAAX; green) were incubated with *L. interrogans* Fiocruz L1-130 (CF640R; cyan) at an MOI of 100. Arrows denote a Rab5a-positive phagosome. Scale bar = 5 μm. Rab5a mean fluorescence intensity (MFI) at was quantified upon peak localization at the phagosome and compared to the entire cytoplasm. Data represents mean±SEM from 10 timelapse images acquired on two separate days. Total duration of each timelapse was 30mins. Statistical significance was calculated by one-way paired t test; **p < 0.01. (C) 91% of the phagocytosed *L. interrogans* observed were subsequently compacted into a spherical vesicle as shown in the top panel. We also observed a fraction (9%) of *L. interrogans* in the cytoplasm escaping the phagosome compaction as shown in the bottom panel. Scale bar = 5 μm. Data are representative of 11 timelapse images acquired on two separate days. (D) Representative immunofluorescence image showing co-localization (arrow heads) of *L. interrogans* Fiocruz L1-130 with Rab5 and LAMP1 in BMDM at 3h post infection. Scale bar = 10 μm. Quantification of co-localization by Mander's coefficient calculated by JACoP plugin in ImageJ. At least three frames per experiment were analyzed with 15-40 cells per frame. Data are pooled from four independent experiments. (C-E) Quantification of IFNβ and TNF by ELISA in culture supernatants of WT BMDMs pretreated with cytochalasin D, bafilomycin A1, and chymostatin, followed by infection with *L. interrogans* (MOI 100) for 6h. Data pooled from three experiments with technical triplicates and expressed as the mean±SEM. Statistical significance was calculated by two-way ANOVA; *p < 0.05, ***p < 0.001, ns = non-significant.

We monitored the fate of internalized *L. interrogans* using time-lapse confocal microscopy. Extracellular *L. interrogans* had an elongated helical morphology characteristic of spirochetes. During internalization, the majority of *L. interrogans* recruited Rab-5 and were compacted into an intracellular vesicle (**Fig 2B**, **2C**). However, in 9% of internalization events observed by time-lapse confocal microscopy, an *L. interrogans* bacteria escaped compaction and could be seen moving through the macrophage cytoplasm, consistent with prior reports that some *L. interrogans* are not digested by the macrophage (**Fig 2C**) [15]. At 3 hours post-infection bacteria were also found to colocalize with LAMP1 in addition to Rab-5 (**Fig 2D**). Together, these data are consistent with phagocytic uptake and processing of bacteria.

To determine if phagocytosis is required for the production of IFNβ, BMDMs were pretreated with cytochalasin D prior to infection with *L. interrogans*. Cytochalasin D treatment resulted in diminished IFNβ secretion (**Fig 2E**). Furthermore, treatment with bafilomycin A1, which inhibits phagosome and lysosomal acidification by blocking the vacuolar ATPase pump [37–39], and chymostatin, an inhibitor of phagosomal proteases [40,41], each resulted in reduced IFNβ following infection with *L. interrogans* (**Fig 2F**). Cytochalasin D, bafilomycin A1, and chymostatin did not affect TNF secretion from BMDM in response to infection with *L. interrogans* (**Fig 2G**). These findings suggest that *L. interrogans* phagocytosis by macrophages is crucial for a robust type I IFN response. Challenge of BMDM with heat-killed *L. interrogans* still resulted in IFNβ secretion indicating that live bacteria were not required for induction of the IFNβ response (**S1B Fig**). The induction of IFNβ secretion was also not dependent on the pathogenicity of the bacteria, as both pathogenic *L. interrogans* Fiocruz L1-130 and saprophytic *L. biflexa* serovar Patoc strain Patoc 1 strains induced similar levels of IFNβ (**S1C Fig**).

## Type I IFN signaling is required for effective control of *L. interrogans in vivo*

Given that *Leptospira* induced a robust type I IFN response, we next assessed if the induction of type I IFN was important for the host's ability to control bacterial replication in a mouse model of infection. C57BL/6J mice infected with *L. interrogans* do not develop fulminant disease; however, *L. interrogans* does chronically colonize the kidneys where they can be shed in the urine [42–44]. To investigate the role of type I IFN signaling in acute *L. interrogans* infection, we used mice deficient in the IFNα/β receptor subunit 1 (IFNAR1). As expected, *in vitro* analysis of BMDMs from *Ifnar1*−/− and WT mice showed no difference in IFNβ secretion following infection with *L. interrogans* (**Fig 3A**); however, the expression of ISGs, such as *Ifit1*, *Ifit3*, *Ifi44*, and *Zbp1* was significantly diminished in *L. interrogans* infected *Ifnar1*−/− BMDMs compared to WT (**Fig 3B**).

Following an intraperitoneal *in vivo* infection with 2x10$^8$ *L. interrogans* (**Fig 3C**), we observed no difference in survival between WT and *Ifnar1*−/− mice (**Fig 3D**). As expected, transcript analysis of *Ifna* and *Ifnb* in the spleen at 24 hours post-infection revealed no differences between *Ifnar1*−/− mice and WT mice (**Fig 3E**). We did however observe higher bacterial

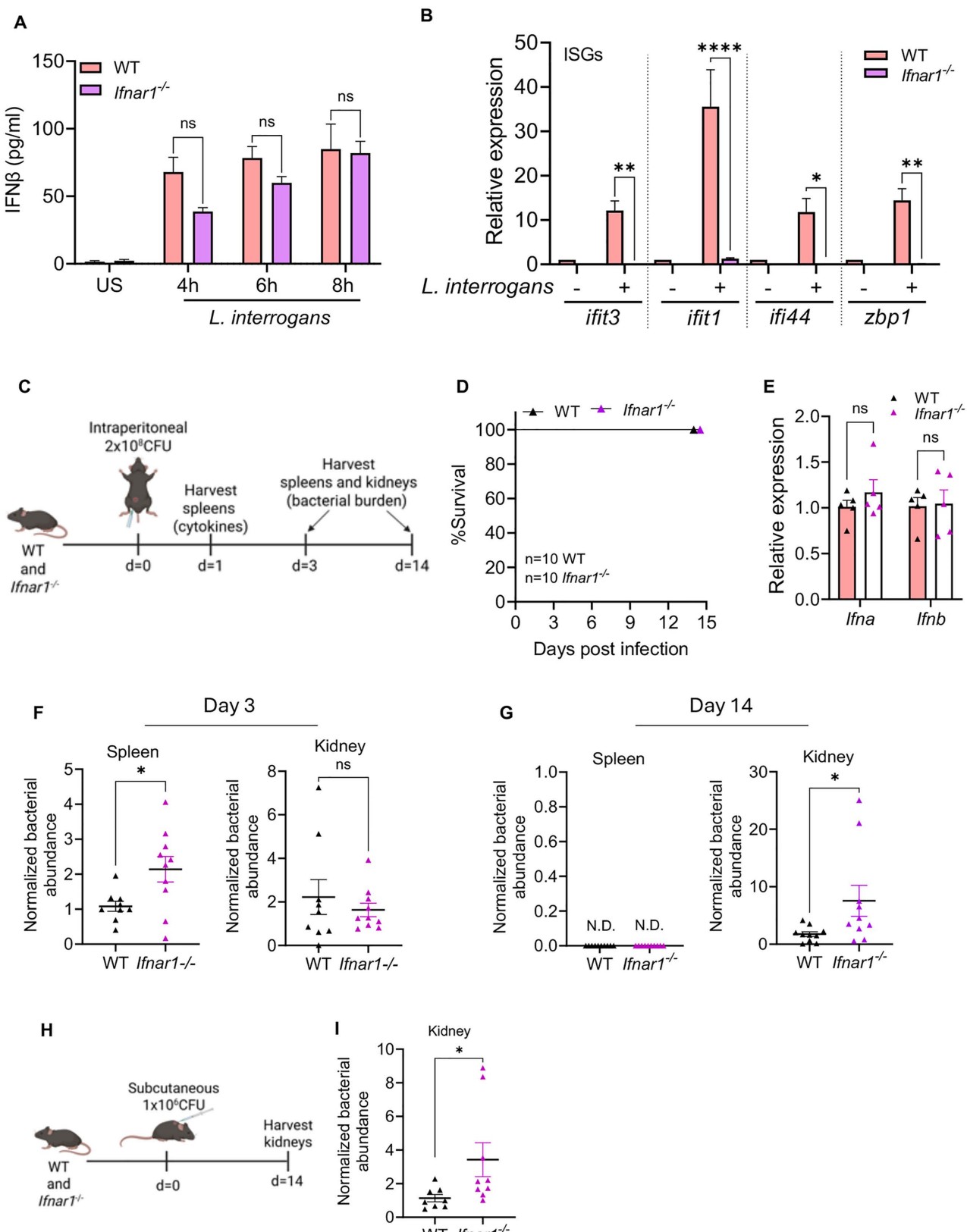

**Fig 3. Increased susceptibility of IFNAR-deficient mice to *L. interrogans* in vivo.** (A) Quantification of IFNβ by ELISA in culture supernatants of WT and Ifnar1-/- BMDMs infected with *L. interrogans* Fiocruz L1-130 (MOI 100) for 4, 6, and 8h. Data are pooled from three experiments and

expressed as the mean ± SEM. US = unstimulated. (B) Relative expression of *ifit1*, *ifit3*, *ifi44*, and *zbp1* transcript levels in WT and *Ifnar1-/-* BMDMs infected with *L. interrogans* at 6h post infection. Data pooled from three experiments and expressed as the mean ± SEM. (C) Schematic of acute infection model showing WT and *Ifnar1-/-* mice were infected intraperitoneally with 2x108 CFU of *L. interrogans* Fiocruz L1-130 and organs harvested either at day 1 post-infection for assessing cytokines or at day 3 and 14 post-infection for assessing bacterial burdens by qPCR. (D) Kaplan-Meier survival curve showing percentage survival of WT and *Ifnar1-/-* mice infected intraperitoneally with 2x108 CFU of *L. interrogans* Fiocruz L1-130 and monitored till day 14 post-infection. Data are pooled from two independent experiments, n = 10 WT, n = 10 *Ifnar1-/-*. (E) Relative expression of *Ifna* and *Ifnb* transcripts as assessed by qPCR in the spleen 24h post-infection. n = 5 WT, n = 5 *Ifnar1-/-*. (F) Bacterial abundance in the spleen and kidney at day 3 post-infection was assessed by qPCR for the *L. interrogans lipL32* gene and normalized to the eukaryotic *PPIA* gene. Data are pooled from two independent experiments: n = 9 WT, n = 10 *Ifnar1-/-*. (G) Bacterial abundance in the spleen and kidney at day 14 post-infection was assessed by qPCR as above. Data are pooled from two independent experiments: n = 10 WT, n = 10 *Ifnar1-/-*. N.D. is nondetectable levels of *lipL32* gene by qPCR. (H, I) Schematic of low dose infection model showing WT and *Ifnar1-/-* mice were infected subcutaneously with 1x106 CFU of *L. interrogans* Fiocruz L1-130 and kidneys were harvested at day 14 post infection. Bacterial abundance was assessed by qPCR for the *L. interrogans lipL32* gene and normalized to the eukaryotic *PPIA* gene. Data are pooled from two independent experiments: n = 8 WT, n = 9 *Ifnar1-/-*. (A, B, E) Statistical significance calculated by two-way ANOVA. (F, G, I) Statistical significance was calculated by Mann-Whitney U test; *p < 0.05, **p < 0.01, ****p < 0.0001, ns = non-significant. Fig 3C and 3H created in BioRender. Gupta, S. (2025) https://urldefense.com/v3/__ https://BioRender.com/u0tcnrs__;!!KOmnBZxC8_2BBQ!zYubkCDZqFuuQ8xxJfSp07zxhKOXmJixzRJBG5gS8wS-X-mEE5q0i2gkPJxzBv5ad2tEsRlfJC-Fy5nQktrnsgU$.

burden in the spleen of *Ifnar1-/-* mice compared to WT at 3 days post-infection (**Fig 3F**). In contrast, no difference in bacterial load was observed in the kidneys at 3 days post-infection (**Fig 3F**). At 14 days post-infection bacterial burdens in the spleen of WT and *Ifnar1-/-* mice was undetectable (**Fig 3G**). Bacterial burden in the kidney of *Ifnar1-/-* mice was increased compared to WT at 14 days post-infection (**Fig 3G**). Together these data demonstrate that *L. interrogans* induces the transcription of ISGs in *L. interrogans* infected macrophages. Furthermore, type I IFN signaling through IFNAR1 is important for bacterial control in the spleen during acute leptospiral infection and also plays a role in controlling chronic kidney colonization. To further confirm that IFNAR1 controls renal colonization by *L. interrogans* we employed a second in vivo model utilizing a low dose subcutaneous infection route that more closely resembles physiologic infection. Mice were infected with 1x10⁶ spirochetes subcutaneously and kidney colonization assessed at 14 days post-infection (**Fig 3H**). *Ifnar1-/-* mice had higher kidney colonization with *L. interrogans* compared to WT mice (**Fig 3I**). Taken together these data suggest that type I IFN signaling through IFNAR plays an important role *in vivo* in limiting chronic kidney colonization.

### *L. interrogans*-induced type I IFN is cGAS-STING dependent

As the type I IFN response from BMDMs in response to *L. interrogans* was dependent on effective bacterial internalization and processing we hypothesized that a cytosolic pattern recognition receptor may be responsible for initiating this response. Additionally, the role of cytosolic DNA sensing in the innate immune response to *L. interrogans* remains unknown. To examine this, BMDMs from mice deficient in cGAS (*Cgas−/−*) and STING (*Sting1gt/gt*) were challenged with *L. interrogans* and *Ifna* and *Ifnb* transcript induction was assessed by qPCR. BMDM from *Cgas−/−* and *Sting1gt/gt* mice showed a marked defect in the transcription of *Ifna* and *Ifnb* at 4 hours post-infection with *L. interrogans* compared to BMDMs from WT mice (**Fig 4A**, **4B**). Consistent with this, BMDMs from *Cgas−/−* and *Sting1gt/gt* mice also had diminished IFNβ secretion, but not TNF secretion, into culture supernatants as measured by ELISA compared to WT macrophages (**Fig 4C**, **4D**). ISGs downstream of IFN signaling were also diminished in BMDMs from *Cgas−/−* and *Sting1gt/gt* mice compared to WT following infection with *L. interrogans* (**Fig 4E**). A similar reduction in *IFNB* transcript levels was observed in the human monocytic THP-1 cell line with *CGAS* and *STING1* gene knockout (**Fig 4F**) indicating *L. interrogans* can activate the cGAS-STING pathway in both mouse and humans. Furthermore, leptospiral DNA directly transfected into BMDMs induced IFNβ production in a cGAS and STING dependent manner (**Fig 4G**).

To confirm that the differences in type I IFN responses seen in the absence of cGAS and STING is not due to differences in phagocytosis pathways we visualized the number of bacteria either bound or internalized by BMDMs. We found similar numbers of bacteria bound to or internalized by WT, *Cgas−/−* and *Sting1gt/gt* BMDMs at 3 hours post-infection

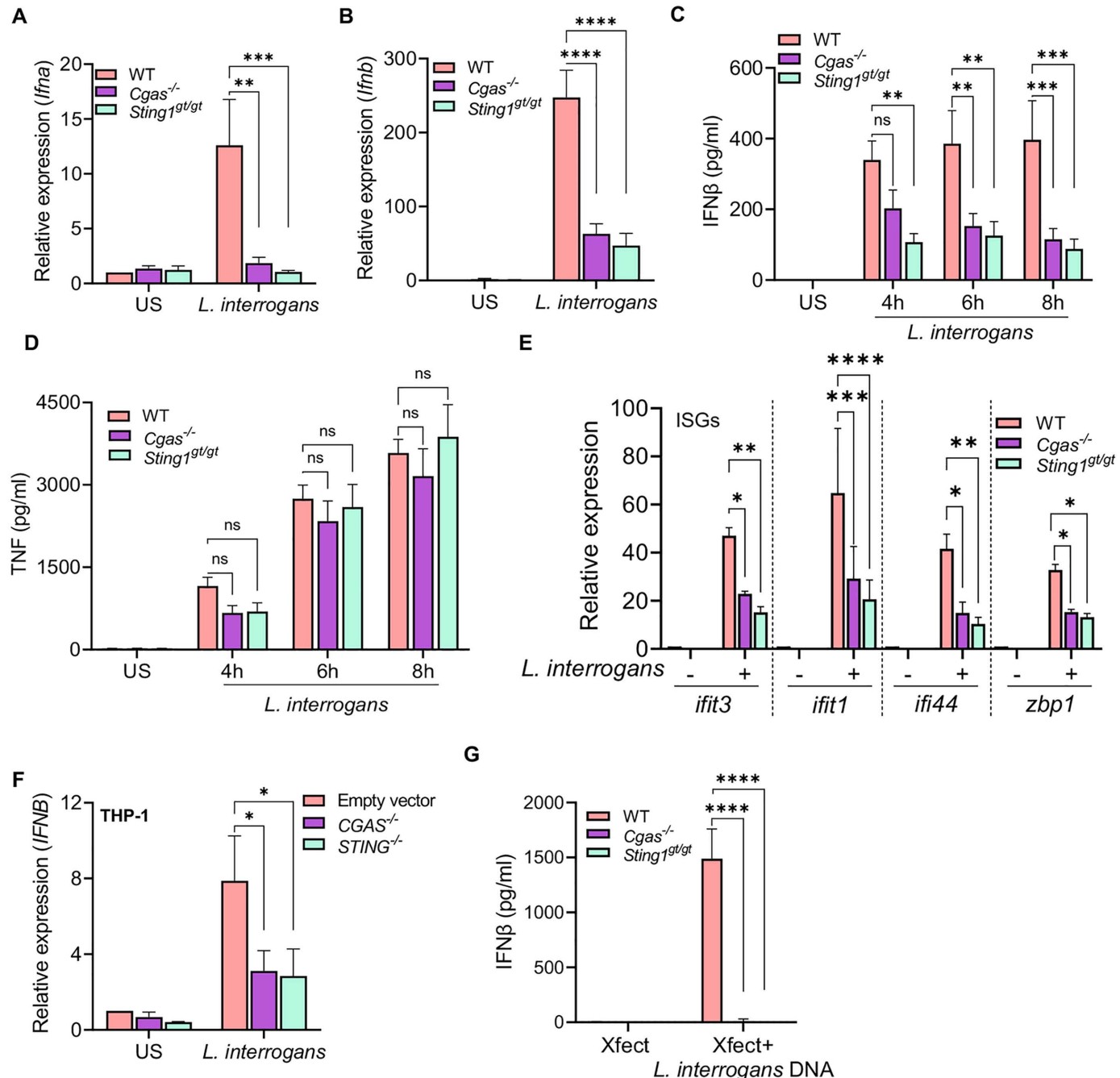

**Fig 4. *L. interrogans* induced type I IFN is dependent on cGAS-STING.** (A, B) Relative expression of *Ifna* and *Ifnb* transcripts assessed by qPCR in BMDMs from WT, *Cgas⁻/⁻*, and *Sting1^{gt/gt}* mice 4h after infection with *L. interrogans* Fiocruz L1-130 (MOI 100). (C, D) Quantification of IFNβ and TNFα by ELISA in culture supernatants of WT, *Cgas⁻/⁻*, and *Sting1^{gt/gt}* BMDMs infected with *L. interrogans* Fiocruz L1-130 (MOI 100) for 4, 6 and 8h. (E) Relative expression of *ifit1*, *ifit3*, *ifi44*, and *zbp1* transcript levels in BMDMs infected with *L. interrogans* (MOI 100) at 6h post-infection. (F) Relative expression of *IFNA* and *IFNB* transcript levels assessed by qPCR in PMA-differentiated human THP-1 cells infected with *L. interrogans* (MOI 100) at 4h post infection. (G) Quantification of IFNβ by ELISA in culture supernatants of WT, *Cgas⁻/⁻*, and *Sting1^{gt/gt}* BMDMs transfected with 500 ng of *L. interrogans* DNA using Xfect transfection agent. (A-G) Data are pooled from three experiments and expressed as the mean ± SEM. Statistical significance was calculated by two-way ANOVA; $*p < 0.05$, $**p < 0.01$, $***p < 0.001$, $****p < 0.0001$, ns = non-significant. US = unstimulated.

(S2 Fig). Taken together these data demonstrate that *L. interrogans* activates the cGAS-STING signaling cascade thereby inducing secretion of type I IFN and the subsequent transcription of ISGs in macrophages. Furthermore, as the production of type I IFN is cGAS dependent, these data suggest that cytosolic recognition of dsDNA initiates this pathway as opposed to bacterial derived cyclic dinucleotides.

**STING is required for effective control of *L. interrogans* in mice *in vivo***

To evaluate the contribution of STING in the control of *L. interrogans in vivo*, we intraperitoneally infected WT and *Sting1^{gt/gt}* mice with *L. interrogans* and assessed survival and bacterial burdens in the spleen and kidneys (**Fig 5A**). No difference in survival was observed between WT and *Sting1^{gt/gt}* mice (**Fig 5B**). Lower expression of *Ifna* and *Ifnb* was seen in the spleen of *Sting1^{gt/gt}* mice compared to WT at 24 hours post-infection (**Fig 5C**). Higher bacterial burdens were observed in the spleen of *Sting1^{gt/gt}* mice compared to WT at 3 days post-infection (**Fig 5D**). No significant difference in bacterial burden was observed in the kidneys between WT and *Sting1^{gt/gt}* mice (**Fig 5D**). At day 14 post-infection bacterial burdens in the spleen of both WT and *Sting1^{gt/gt}* mice was undetectable (**Fig 5E**). However, bacterial burdens in the kidney of *Sting1^{gt/gt}* mice was increased compared to WT at day 14 post-infection (**Fig 5E**). To confirm that STING played a role in controlling kidney colonization by *L. interrogans* we used a second *in vivo* infection model in which WT and *Sting1^{gt/gt}* mice were subcutaneously infected with $1\times10^6$ spirochetes and kidney colonization assessed at 14 days post-infection (**Fig 5F**). *Sting1^{gt/gt}* mice had higher kidney colonization with *L. interrogans* compared to WT mice (**Fig 5G**). Taken together these data suggest that cytosolic recognition of *L. interrogans* activates the cGAS-STING pathway to control *L. interrogans* replication *in vivo* and limits chronic kidney colonization.

## Discussion

In this work, we have shown that *L. interrogans* induces a robust type I IFN response from human and mouse macrophages. The induction of type I IFN by *L. interrogans* was dependent on the cGAS-STING pathway as macrophages deficient in either cGAS or STING had markedly diminished expression of type I IFN and ISGs. Interestingly, the *L. interrogans* mediated production of type I IFN required bacterial phagocytosis and phagosomal acidification. We further demonstrate that the induction of type I IFN plays an important role in the host response to infection with *L. interrogans in vivo*; mice deficient in STING or IFNAR displayed increased susceptibility to *L. interrogans* infection and had increased kidney colonization at 14 days post-infection.

Leptospires are largely thought to be extracellular pathogens; however, studies have shown leptospires to be found within macrophages [13,15]. The fate and impact of these intracellular leptospires has been unclear [7]. Santecchia et al. demonstrated that *L. interrogans* can actively enter into both mouse and human macrophages [15]. Surprisingly, they also found that the leptospires did not replicate within macrophages but could rapidly exit these cells intact [15]. Consistent with this report, our time-lapse confocal microscopy studies documented occasional leptospiral escape from compaction within a subcellular compartment (**Fig 2C**). Using a pH-sensitive dye to label *L. interrogans*, we found that leptospires were being taken up in acidic sub-cellular compartments in macrophages. A fraction of internalized leptospires colocalized with Rab5 and LAMP1, suggesting the presence of leptospires in phagocytic vesicles with progression of phagosomal maturation. We further found that the pharmacologic inhibitors cytochalasin D, bafilomycin A and chymostatin, that target phagocytic uptake, phagosomal acidification, and proteolytic activity, respectively, inhibited *L. interrogans* induced IFNβ production. These data suggest that the phagocytosis and phagosomal processing of *L. interrogans* is required for the activation of the cGAS-STING pathway. It remains unclear to what extent these internalized leptospires are killed or damaged resulting in bacterial DNA release into the macrophage cytosol.

The cGAS-STING pathway can drive the production of type I IFN in response to pathogen-derived DNA as well as endogenous host DNA [45–48]. cGAS can bind to dsDNA and produce cGAMP, which then binds to and activates STING on the endoplasmic reticulum membrane [46,48–50]. This results in TBK1 phosphorylation, which phosphorylates the

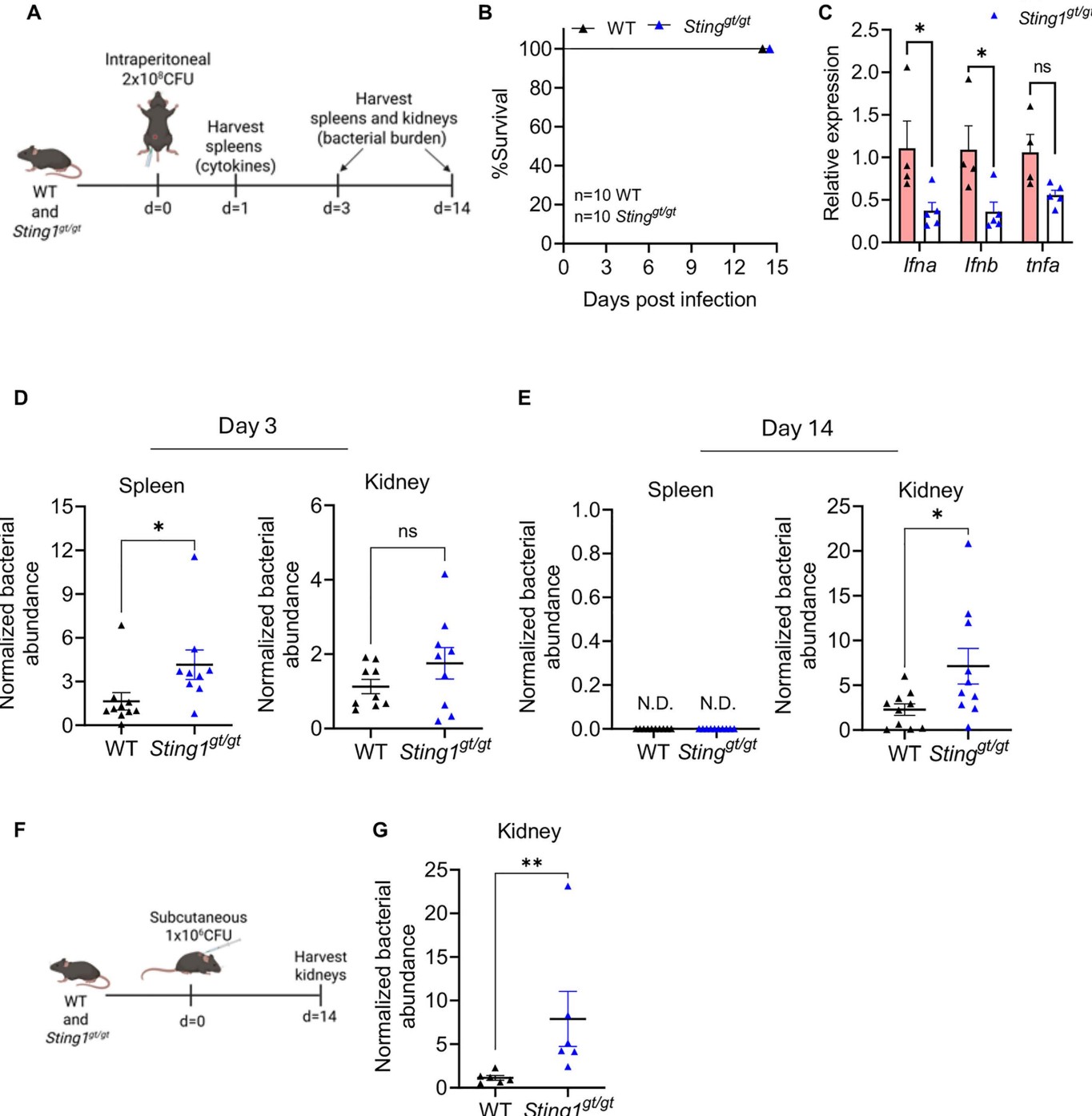

**Fig 5. Increased susceptibility of STING-deficient mice to _L. interrogans in vivo_.** (A) Schematic of acute infection model showing WT and _Sting1gt/gt_ mice were infected intraperitoneally with 2x10⁸ CFU of _L. interrogans_ Fiocruz L1-130 and organs harvested either at day 1 for assessing cytokines or at day 3 and 14 for assessing bacterial burdens by qPCR. (B) Kaplan-Meier survival curve showing percentage survival of WT and _Sting1gt/gt_ mice infected intraperitoneally with 2x10⁸ CFU of _L. interrogans_ Fiocruz L1-130 and monitored till day 14 post-infection. Data are pooled from two independent experiments, n = 10 WT, n = 10 _Sting1gt/gt_. (C) Relative expression of _Ifna_, _Ifnb_ and _tnfa_ transcripts as assessed by qPCR in spleen 24h post-acute infection in WT and _Sting1gt/gt_ mice. n = 4 WT, n = 5 _Sting1gt/gt_. (D) Bacterial abundance in the spleen and kidney at day 3 post-infection was assessed by qPCR for the _L. interrogans lipL32_ gene and normalized to the eukaryotic _PPIA_ gene. Data are

pooled from two independent experiments: n = 10 WT, n = 9 *Sting1gt/gt*. (E) Bacterial abundance in the spleen and kidney at day 14 post-infection was assessed by qPCR as above. Data are pooled from two independent experiments: n = 10 WT, n = 10 *Sting1gt/gt*. N.D. is nondetectable levels of *lipL32* gene by qPCR. (F, G) Schematic of low dose infection model showing WT and *Stinggt/gt* mice were infected subcutaneously with 1x10⁶ CFU of *L. interrogans* Fiocruz L1-130 and kidneys were harvested at day 14 post infection. Bacterial abundance was assessed by qPCR for the *L. interrogans lipL32* gene and normalized to the eukaryotic *PPIA* gene. Data are pooled from two independent experiments: n = 6 WT, n = 6 *Stinggt/gt*. (C) Statistical significance was calculated by two-way ANOVA. (D, E, G) Statistical significance calculated by Mann-Whitney U test; *p < 0.05, **p < 0.01, ns = non-significant. Fig 5A and 5F created in BioRender. Gupta, S. (2025) https://urldefense.com/v3/__ https://BioRender.com/u0tcnrs__;!!KOmnBZxC8_2BBQ!zYubkCDZqFuuQ8xxJfSp07zxhKOXmJixzRJBG5gS8wS-X-mEE5q0i2gkPJxzBv5ad2tEsRlfJC-Fy5nQktrnsgU$.

transcription factor IRF3 and drives the induction of type I IFN [51–53]. We observed that *Leptospira* induces a type I IFN response, coinciding with the phosphorylation of TBK1 and IRF3, supporting a role for cGAS-STING in the detection of intracellular *L. interrogans*. Several intracellular bacterial pathogens, including *Salmonella enterica* serovar Typhimurium, *L. monocytogenes,* and *M. tuberculosis* have been shown to activate the cGAS-STING pathway to drive type I IFN production [17,19,54]. Extracellular bacteria, including *Klebsiella pneumoniae* and *Pseudomonas aeruginosa*, have also been shown to activate the cGAS-STING pathway [20]. The spirochete *B. burgdorferi* can also activate the cGAS-STING pathway in macrophages and fibroblasts [23]. Although cGAS-STING did not appear to play a role in control of *B. burgdorferi* burdens *in vivo*, inflammation and joint pathology were diminished in cGAS-deficient mice infected with *B. burgdorferi* [23]. The mechanism by which cGAS-STING recognizes *L. interrogans* remains unclear. Bacterial derived cyclic dinucleotides have been shown to directly activate STING [24–26]. Although *L. interrogans* can produce cyclic dinucleotides [28,29], our finding that cGAS is required for *L. interrogans*-driven type I IFN production suggests that sensing of cytosolic DNA is required to engage the cGAS-STING pathway in response to *L. interrogans* and that *L. interrogans* derived cyclic dinucleotides do not directly activate STING. As internalization and phagocytic processing of *L. interrogans* is required for type I IFN production by macrophages it is possible that some internalized spirochetes either shed DNA or are damaged within the phagosome resulting in release of bacterial DNA that can be recognized by cytosolic cGAS. Another possibility is that infection of macrophages with *L. interrogans* results in mitochondrial stress resulting in the release of mtDNA which can then activate cGAS-STING [54]. It should be noted that other pattern recognition receptors, including RIG-I/MDA5/MAVS, TLR3, TLR4, TLR7, TLR8, and TLR9, may be involved in the induction of type I IFN responses [55–62]. The role of these pattern recognition receptors in driving the type I IFN response to *L. interrogans* will require further investigation.

The murine model of leptospirosis remains poorly characterized, leaving many questions unanswered regarding the host response to *Leptospira* spp. A recent study by Papadopoulos et al. on the progression of acute leptospirosis in mouse models provides valuable insights into how severe leptospirosis in mice mimics aspects of human disease [63]. This study attributes the observed mouse mortality in the acute leptospirosis model to severe myocarditis and neutrophil infiltration, in the absence of a significant cytokine storm [63]. Our *in vivo* infection of *Sting1gt/gt* and *Ifnar1-/-* mice with *L. interrogans* revealed a role for type I IFN in both the early control of bacterial replication as well as in chronic kidney colonization with *L. interrogans*. The effect of the *L. interrogans*-driven type I IFN response on neutrophil function in vivo remains to be evaluated. In addition, the contribution of other cell types, such as renal tubular epithelial cells, in the production of type I IFN in response to *L. interrogans* will be important to determine.

In conclusion, our study is the first to show that *L. interrogans* activates the cytosolic cGAS-STING DNA sensing pathway in mouse and human macrophages resulting in a type I IFN response. Future studies are required to determine the source of cytosolic DNA that is recognized by cGAS-STING following *L. interrogans* infection of macrophages. Additionally, determining the intracellular fate of *L. interrogans* will be important in understanding if bacterial damage and release of bacterial DNA is a prerequisite for activation of cGAS-STING. We also found that activation of STING and signaling through IFNAR are required for controlling *L. interrogans* replication and kidney colonization in mice in vivo. Evaluating

the role of type I IFN production during the course of human disease remains an unresolved question and may provide insights into species specific responses.

## Materials and methods

### Ethics statement

All studies with mice were performed in accordance with the recommendations in the Guide for the Care and Use of Laboratory Animals of the National Institutes of Health and were reviewed and approved by the Institutional Animal Care committee at Cedars-Sinai Medical Center (CSMC IACUC #006777).

### Mice

Wild-type (WT) C57BL/6J (JAX stock# 000664) mice were purchased from Jackson laboratories and used as controls unless otherwise stated. The generation of *Sting1*$^{gt/gt}$ (JAX stock# 017537), *Cgas*$^{-/-}$ (JAX stock# 026554) and *Ifnar1*$^{-/-}$ (JAX stock# 028288) have been previously described [64–66]. Mice were bred and maintained in a specific-pathogen free facility. Both male and female mice 6–12 wk of age were used; however, mice were sex and age matched for individual experiments.

### Bacterial culture

*Leptospira interrogans* serovar Copenhageni strain Fiocruz L1-130 and *Leptospira biflexa* serovar Patoc strain Patoc 1 were grown in Hornsby-Alt-Nally (HAN) media [67]. Cultures were incubated at 30°C and passaged weekly. *L. interrogans* was passaged a maximum of five times after collection from kidneys of infected Golden Syrian hamsters. Culture densities were determined by diluting 10 μL of culture into phosphate-buffered saline (PBS) and counting by darkfield microscopy with a Zeiss Axio Lab A1 or AmScope B340 Series microscope. Prior to experiments, leptospires were centrifuged at 9,000 x g for 4 min at room temperature. For *in vitro* experiments, leptospires were resuspended in DMEM to a final density of $1 \times 10^9$/mL. For heat killing, the leptospires were heated at 56°C for 30 min. For labelling of bacteria with pHrodo Red, succinimidyl ester (Invitrogen # P36600), leptospires were incubated for 45 min at room temperature with 0.1 mM dye. Labelled bacteria were washed and then incubated with BMDMs to analyze increase in fluorescence using an Incucyte SX5 Live-Cell Analysis System (Sartorius). For *in vivo* experiments, leptospires were resuspended in HAN media to a density of $1 \times 10^9$/mL following centrifugation of the culture. After resuspension, 200 μL was used to inoculate mice for acute infection; for renal colonization, the cell suspension was diluted 100-fold prior to inoculation of mice with 100 μL.

### *In vitro* stimulation of bone marrow–derived macrophages

Bone marrow–derived macrophages (BMDMs) were generated from cells collected by flushing mouse femurs and tibias and culturing the cells in 10 mm dish for 6–7 days in DMEM (Corning #10–013-CV) with 10% FBS (R&D systems #S12450H), penicillin-streptomycin (Gibco # 15140122), and L-glutamine (Gibco #25030–081) in the presence of 20% L929 cell-conditioned medium. Upon differentiation, BMDMs were harvested using Versene (Gibco# 15040066) and used for subsequent experiments. BMDMs were either left unstimulated or infected with *L. interrogans* Fiocruz L1-130 or *L. biflexa* Patoc I at multiplicity of infection (MOI) of 100 for 4, 6, and 8 h. Wherever indicated, BMDMs were pre-incubated for 45–60 mins with 2 μM cytochalasin D (Sigma-Aldrich #C8273), 400 nM bafilomycin A (Invivogen #tlrl-baf1) and 100 μM chymostatin (Cayman Chemical #15114) before addition of bacteria. For transfection of leptospiral DNA, 500 ng of DNA isolated from *L. interrogans* Fiocruz L1-130 cultures was incubated with Xfect transfection reagent (Takara Bio #631317) for 10 min and BMDMs were treated with the mixture for 6 h. After respective stimulations, supernatants were collected for quantification of cytokines and cells were preserved in RNAlater Stabilization Solution (Invitrogen #AM7020) for RNA isolation using RNeasy Mini kit (Qiagen #74106) as per the manufacturers' instruction.

Cytokines and chemokines were quantified in the cell culture supernatants using mouse ELISA kit for TNF (Invitrogen #88-7324-88), IFNβ (R&D Systems #DY8234), IL-10 (R&D Systems #DY417) and IL-6 (R&D Systems #DY406) following the manufacturers' protocol.

## Lentivirus production and infection

Rab5a was expressed using lentiviral infection. Lenti-X 293T cells were transfected with pMD2.g (gift from Didier Trono, Ecole Polytechnique Federal de Lausanne (EPFL), Lausanne, Switzerland, Addgene plasmid #12259) containing VSV-G envelope protein, pCMV-dR8.2 (gift from Bob Weinberg, MIT, Cambridge, MA, USA, Addgene plasmid #8455), and pHR-Rab5a-mCherry (gift from Ivan Yudushkin, Medical University of Vienna, Austria, plasmid #72901; Addgene [68] or pHR-CAAX-GFP [69] using lipofectamine LTX (cat #15338–100; Invitrogen). The media was harvested 72 h after transfection, filtered through a 0.45 - μm filer (cat #SLHVM33RS; Millipore) and concentrated using LentiX (cat #631232; Takara Biosciences). Concentrated lentivirus was added to BMDMs on day 2 of differentiation, and cells were used for experiments between days 7–10.

## Time-lapse confocal microscopy

$5x10^4$ BMDMs were plated per well of a 96 well glass-bottom MatriPlate (cat #MGB096-1-2-LG-L; Brooks) 24 h prior to experimentation. *L. interrogans* Fiocruz L1-130 were labeled with 0.1mM CF640R-succinimydyl ester (SE) (Biotium # 92108) for 1 h. Labeled *L. interrogans* Fiocruz L1-130 were added to BMDMs at an MOI of 100 and spun down at 5,000 x g for 3 min to initiate contact with macrophages. Images were acquired on a spinning disc confocal microscope (Nikon Ti2-E inverted microscope with a Yokogawa CSU-W1 spinning disk unit and a Hamamatsu Orca Fusion BT scMos camera) equipped with a 100 x 1.49 NA oil immersion objective. The microscope is also fitted with a piezo Z drive. An OkoLabs stage top incubator was used to maintain a temperature of 37°C and CO2 level of 5%. Image acquisition was controlled using Nikon Elements. Rab5a recruitment to internalized leptospires was quantified in ImageJ by measuring the mean fluorescent intensity of an ROI immediately surrounding the bacteria. Cytoplasmic Rab5a intensity was measured for an ROI created by tracing the cell membrane and subtracting the nuclear area. Background fluorescence was measured immediately outside of each cell and subtracted from the phagosomal and cytoplasmic values. Leptospira were analyzed only if we could visualize internalization of a living *L. interrogans* Fiocruz L1-130 and tracking the phagosome for 10 min.

## Immunofluorescence staining and confocal microscopy

$3x10^4$ BMDMs were seeded in an 8-well Millicell EZ Slide (Millipore Sigma #PEZGS0816) and incubated for 3 h with *L. interrogans* Fiocruz L1-130 at MOI 100 or left unstimulated. Cells were washed with PBS three times and fixed with 4% PFA (Electron Microscopy Sciences #15710) for 10 min at room temperature. For analyzing bound bacteria, after fixation, cells were not permeabilized and proceeded with immunofluorescence (IF) staining. For analyzing total bound and internalized bacteria, fixed cells were permeabilized using 0.1% TritonX-100 (VWR Life Science # 9002-93-1) for 10 min. Cells were then incubated in IF blocking buffer containing 2.5% BSA (RPI # A30075) and normal donkey serum (Abcam #ab7475) in PBS (Corning #21–040-CV) for 1 h at room temperature. Primary antibodies: rat anti-mouse LAMP1 (clone 1D4B, eBioscience), mouse anti-mouse Rab5 (clone rab5–65, Sigma-Aldrich), and rabbit anti-sera raised against *L. interrogans* Fiocruz L1-130 were diluted in IF blocking buffer and added to cells for overnight incubation at 4°C. Cells were washed three times with PBS and AF488 anti-rabbit IgG, AF555 anti-mouse IgG, and AF647 anti-rat IgG diluted in IF buffer were incubated with the cells for 1 h at room temperature. Cells were washed three times with PBS before removing the well borders. VECTASHIELD Vibrance Antifade Mounting Medium with DAPI (Vector Laboratories #H-1800) was added on top of each well of stained cells and sealed with a coverslip. Imaging was done using either Zeiss Z780 confocal microscope or Leica Stellaris confocal microscope. For counting the number of bacterial puncta that are bound vs the total

number of bacteria bound and internalized, the Image J manual cell counting tool was used. Three to four frames per well per experiment were counted with each frame having 50–150 cells. For analyzing the percentage of total bacteria that colocalizes with Rab5 or LAMP1, the JaCoP pixel-based colocalization tool was used to calculate the Manders' coefficient using manual thresholding. Three to four frames per well per experiment were analyzed for colocalization with each frame having 15–40 cells.

### *In vivo* infections

For acute infection, mice were injected intraperitoneally with $2\times10^8$ bacteria and on day 3 and 14 post-infection spleen and kidney were harvested and stored in RNAlater Stabilization Solution (Invitrogen #AM7020) for further processing. Tissues were weighed and homogenized in PBS with 2.8 mm ceramic beads (Omni #19–646) using a FastPrep-24 Classic bead beating grinder and lysis system (MP #116004500). Subsequently DNA was extracted from appropriate homogenized tissue amount using DNeasy Blood and Tissue kit (Qiagen # 69506) using the manufacturers' protocol. RNA was extracted from homogenized tissues for assessing cytokine levels using RNeasy Mini kit (Qiagen #74106) as per the manufacturers' instruction. To assess kidney colonization, mice were injected subcutaneously with $1\times10^6$ bacteria and on day 14 following infection the kidney was harvested and processed for DNA isolation as above.

### qPCR

cDNA was synthesized from the isolated RNA using PrimeScript RT master mix (Takara Bio #RR036A) as per the manufacturers' protocol. Transcripts were amplified with the primers listed in S1 Table using PowerUP SYBR Green PCR master mix (Applied Biosystems #25742) in Applied biosystems ViiA7 96 well system. The transcript levels were normalized to housekeeping gene β-actin and data are shown as fold change of transcripts relative to mean of WT group using the ΔΔ threshold cycle (Ct) method. For quantifying the relative abundance of bacteria in kidneys and spleen of infected mice, the DNA was amplified with primers specific to the *lipL32* gene for *L. interrogans* and normalized to eukaryotic tissue inputs by amplification of *PPIA* housekeeping gene. Data are shown as relative abundance of bacteria normalized to tissue inputs and compared to the WT group on day 3 or day 14 post-infection using the ΔΔ Ct method.

### Cell lines

Human monocyte cell line THP-1 with *CGAS* and *STING1* gene knockout were generated by CRISPR/Cas9 as previously described [70]. THP-1 monocytes were grown in RPMI-1640 medium (ATCC #30–2001) with 10% FBS (R&D systems #S12450H), penicillin-streptomycin (Gibco # 15140122), L-glutamine (Gibco #25030–081) and β-mercaptoethanol (Thermo Fisher # 21985023). THP-1 monocytes were differentiated using 50 nM PMA for 3–4 h, followed by an overnight rest period. The cells were then infected with *L. interrogans* Fiocruz L1-130 at MOI 100; 4 h later the cells were harvested for RNA isolation.

### Western blotting

Cell lysates were prepared in radioimmunoprecipitation assay (RIPA) lysis buffer (Cell Signaling Technology #9806S) with Halt Protease and Phosphatase Inhibitor Cocktail (Thermo scientific #78440). Proteins were separated on NuPAGE Bis-Tris Mini Protein Gels, 4–12%, 1.0–1.5 mm and transferred to a polyvinylidene difluoride (PVDF) membrane using either the XCell II blotting system (Invitrogen #EI0002) or iBlot 2 Transfer system (Invitrogen #IB24002). Membranes were blocked with 1% BSA in tris-buffered saline (TBS) with 0.1% Tween-20 and incubated with primary antibody overnight at 4°C. Primary antibodies from Cell signaling technologies (CST) were used as follows: TBK1 (CST #3504S), p-TBK1 (CST #5483S), IRF3 (CST #4302S), p-IRF3 (CST #4947S) and β-actin (CST #4970S). After washing, membranes were incubated with HRP-tagged anti-rabbit IgG (Cell Signaling Technology #7074S). Membranes were developed using SuperSignal West Pico PLUS (Thermo Scientific #34580) or Femto substrate (Thermo Scientific #34094) and imaged in LI-COR Odyssey Fc.

## Statistics

Data were graphed and the indicated statistical tests performed using GraphPad Prism 9 software. qPCR experiments were done in technical triplicates; unless otherwise specified results were pooled from three biological replicates and expressed as the mean±SEM. ELISAs were performed as either technical duplicates or triplicates; unless otherwise specified results were pooled from three biological replicates and expressed as the mean±SEM. If not otherwise stated, analysis was performed with Unpaired *t* test with Welch's correction for single comparison or Two-way ANOVA for multiple comparisons.

## Supporting information

**S1 Fig. Analysis of IFNβ production from BMDM infected with *Leptospira*.** (A) Dose dependent induction of IFNβ was quantified by ELISA in culture supernatants of WT BMDMs infected with *L. interrogans* (MOI 1, 10, 100) for 4, 6, and 8h. Data are pooled from two independent experiments and expressed as the mean±SD. (B) Quantification of IFNβ levels by ELISA in culture supernatants of WT BMDM infected with live or heat-killed *L. interrogans* (MOI 100) at 6h post-infection. Data are pooled from three independent experiments. Statistical significance was calculated by unpaired Student's *t* test. (C) Quantification of IFNβ levels by ELISA in culture supernatants of WT BMDM infected with *L. interrogans* Fiocruz L1-130 (MOI 100) or *L. biflexa* Patoc 1 (MOI 100) at 4, 6, and 8h post-infection. Statistical significance calculated by two-way ANOVA. ns = non-significant.
(DOCX)

**S2 Fig. Analysis of *L. interrogans* binding and internalization by BMDM.** Quantification of immunofluorescence images of bacteria bound or total bacteria (bound and internalized) in WT, *Cgas*[-/-] and *Sting1*[gt/gt] BMDMs at 3h post infection with *L. interrogans* (MOI 100). Data are pooled from three frames per experiment from three independent experiments. Statistical significance was calculated by two-way ANOVA. ns = non-significant.
(DOCX)

**S1 Table. Primers used for qPCR.**
(DOCX)

## Author contributions

**Conceptualization:** Suman Gupta, Suzanne L. Cassel, David A. Haake, Fayyaz S. Sutterwala.

**Data curation:** Suman Gupta.

**Formal analysis:** Suman Gupta.

**Funding acquisition:** Jargalsaikhan Dagvadorj, Meghan A. Morrissey, Suzanne L. Cassel, David A. Haake, Fayyaz S. Sutterwala.

**Investigation:** Suman Gupta, James Matsunaga, Bridget Ratitong, Andrew Manion, Sana Ismaeel, Diogo G. Valadares, Andrea J. Wolf.

**Methodology:** Suman Gupta, James Matsunaga, Bridget Ratitong, Andrew Manion, Diogo G. Valadares, Jargalsaikhan Dagvadorj, Jenifer Coburn, Andrea J. Wolf, Meghan A. Morrissey, Suzanne L. Cassel, David A. Haake, Fayyaz S. Sutterwala.

**Resources:** A. Phillip West, Nagaraj Kerur, Christian Stehlik, Andrea Dorfleutner, David A. Haake.

**Supervision:** Meghan A. Morrissey, Suzanne L. Cassel, David A. Haake, Fayyaz S. Sutterwala.

**Writing – original draft:** Suman Gupta, David A. Haake, Fayyaz S. Sutterwala.

**Writing – review & editing:** Suman Gupta, Andrew Manion, Christian Stehlik, Andrea Dorfleutner, Jenifer Coburn, Meghan A. Morrissey, Suzanne L. Cassel, David A. Haake, Fayyaz S. Sutterwala.

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
