## [Decision Letter · Decision Letter 0]

30 Jul 2025

cGAS-STING dependent type I IFN protects against Leptospira interrogans renal colonization in mice.

PLOS Pathogens

Dear Dr. Sutterwala,

Thank you for submitting your manuscript to PLOS Pathogens. After careful consideration, we feel that it has merit but does not fully meet PLOS Pathogens's publication criteria as it currently stands. Therefore, we invite you to submit a revised version of the manuscript that addresses the points raised during the review process.

Please submit your revised manuscript within 60 days Sep 28 2025 11:59PM. If you will need more time than this to complete your revisions, please reply to this message or contact the journal office at plospathogens@plos.org. Please include the following items when submitting your revised manuscript:

We look forward to receiving your revised manuscript.

Kind regards,

Dana J. Philpott

Academic Editor

PLOS Pathogens

Thomas Guillard

Section Editor

PLOS Pathogens

Editor-in-Chief

PLOS Pathogens

orcid.org/0000-0003-2946-9497

Editor-in-Chief

PLOS Pathogens

orcid.org/0000-0002-7699-2064

**Additional Editor Comments:**

Thank you for submitting your work to PLOS Pathogens. While the work is generally well done and the findings are of interest, the reviewers had a number of concerns that will need to be addressed. For one, the authors need to clarify some of the experimental details. For instance, with regards to replicates, Figure 1 and 2 state that “Data pooled from three experiments” – how many technical replicates would this be per experiment? Furthermore, some of the findings do not support the conclusions made by the authors and it would be important to moderate some of their assertions.

**Journal Requirements:**

At this stage, the following Authors/Authors require contributions: Suman Gupta, James Matsunaga, Bridget Ratitong, Sana Ismaeel, Diogo Valadares, Phillip West, Nagaraj Kerur, Christian Stehlik, Andrea Dorfleutner, Jargalsaikhan Dagvadorj, Jenifer Coburn, Andrea Wolf, Suzanne Cassel, David Haake, and Fayyaz S. Sutterwala. Please ensure that the full contributions of each author are acknowledged in the "Add/Edit/Remove Authors" section of our submission form.

4) We notice that your supplementary Figures, and Tables are included in the manuscript file. Please remove them and upload them with the file type 'Supporting Information'. Please ensure that each Supporting Information file has a legend listed in the manuscript after the references list.

Potential Copyright Issues:

- Figures 3 and 5.. Please confirm whether you drew the images / clip-art within the figure panels by hand. If you did not draw the images, please provide (a) a link to the source of the images or icons and their license / terms of use; or (b) written permission from the copyright holder to publish the images or icons under our CC BY 4.0 license. Alternatively, you may replace the images with open source alternatives. See these open source resources you may use to replace images / clip-art:

**Reviewers' Comments:**

Reviewer's Responses to Questions

**Part I - Summary**

Reviewer #1: The manuscript by Gupta et al. describe in vivo and in vitro experiments to demonstrate that cGAS-STING-driven type I IFN is required for the host defense against L. interrogans. Despite the interesting results showing a potential role of this pathway on overall leptospirosis infection, the methods and results presented here are not sufficient to eliminate the bias and do not support the conclusions of the authors. This pathway might have a role on the innate immune response against leptospires, but the results presented here are not enough to show that they are essential or even important to control leptospiral infection.

Reviewer #2: This manuscript presents a novel approach to targeting a major innate immune signaling pathway—cGAS-STING—which is central to sensing extracellular dsDNA and triggering downstream type I interferon responses. While the study is well-designed and employs a variety of approaches, the current experimental evidence partially supports the major claims. Additional experiments—particularly involving cGAS-deficient models, increasing biological replicates, and further mechanistic insights would further substantiate the conclusions.

Reviewer #3: This manuscript presents a investigation into the role of the cGAS-STING signaling pathway in the host response to Leptospira interrogans infection. The authors demonstrate that Leptospira spp., although traditionally classified as extracellular bacteria, can be detected in the cytosol and within phagolysosomal compartments of macrophages. This intracellular presence allows for the sensing of leptospiral DNA by cGAS, followed by activation of the STING pathway and subsequent induction of type I interferon (IFN-I) responses. Using both in vitro infection of macrophages and in vivo infection models in C57BL/6 mice, the study shows that deficiency of cGAS or STING results in diminished IFN-I production, increased bacterial load, and enhanced renal colonization.

The experimental design is solid, and the data convincingly support the authors' conclusions. The manuscript is clearly written and well-organized. Overall, this is an important and well-executed study.

**Part II – Major Issues: Key Experiments Required for Acceptance**

Reviewer #1: The conclusion of the authors is biased and without experimental foundation. Assuming that the pathways that were studies here are important for the host response, you would expect a change in outcomes. Other studies have shown, for example, that a dose of 2x10◯8 leptospires can cause death in mouse, specifically C57BL/6J. One would expect that with such important role and such a high dose of bacteria, knockout animals would become clinically ill or die. The difference in number of leptospires in the tissue is not enough to conclude that this pathways is “important” to control host infection. Figure 3H shows only a couple of animals that are outliers, with higher number of leptospires compared to wt animals. If there was an important role, you would expect to see the majority of the animals with higher number of leptospires in their kidneys.

It is not clear why two different doses were used to infect animals for specific outcomes. It seems that both doses lead to colonization, so the choice to have two doses makes no sense, especially since a higher dose lead to no difference in leptospires in the kidneys, for example.

In vitro experiments would be better interpreted if a non-pathogenic leptospires was used as control. It is unclear if all the effects seen here are due to pathogenic leptospires or simply leptospires in general, and that would be essential to take in consideration the conclusions and discussion established by the authors.

Reviewer #2: Figures 2 are based on three biological replicates, which is insufficient to robustly support the claims regarding phagolysosomal trafficking kinetics in BMDMs during L. interrogans infection. This experiment needs to be repeated. In Fig 2A Cyto D does not show error bars. Moreover, cytochalasin D treatment data were not provided on IFN-α, and there is a lack of experimental validation for actin polymerization (e.g., phalloidin staining). It would strengthen the study to include additional markers such as p65, Rab7, LAMP2, ORP, and LC3 to assess phagolysosomal and autophagic processes.

Results Fig 3. The description of these results needs to be clarified (add time points; Line 178: clarify ISGs, why is this expected), was IFN-alpha protein not tested? were ISGs mRNA tested at 4h and 8h? if not significant state so in the results and show data in supplemental material. The figures need to represent clearly what was done (protein detection by ELISA, mRNA qPCR etc). Fig 3F, it is unclear why the mRNA of ifna and ifnb is tested at 24h when the mice were euthanized at 3 dpi; the rationale for using different infection doses, animal numbers and routes between day 3 and day 14 experiments is unclear. Additionally, bacterial burden in the spleen, or IFN-α/β expression data at day 14, are not shown and would be important to support the conclusions. How this data supports the current claim of protecting kidney colonization needs to be clarified.

Results Fig 4. Was IFN-alpha protein not tested (4C and 4G? If no differences the data should be reported in this section and shown in supplemental material.

Results Fig 5. The comments made above to Fig 3 apply to Fig 5. No significant differences were observed between wild-type and STING-deficient mice at day 3 post-infection. Although some differences emerge at day 14, the data are limited and insufficient to substantiate the key claims. It is unclear why both spleen and kidney burdens are shown at day 3 but only kidney is assessed at day 14, and why different infection strategies were used across these timepoints. A clearer rationale is needed to justify expected differences at d3 and d14.

Figure S1 does not show significant differences in IFN induction between pathogenic versus saprophytic Leptospira, or between live and heat-killed L. interrogans. These findings need to be better explained since the central claim that type I IFN signaling is specifically activated by pathogenic L. interrogans via the cGAS-STING pathway.

While the burden of Leptospira is higher in the kidneys of subcutaneous infected mice in the knockout (Sting1 and Ifnar1), the WT mice still have a substantial amount of Leptospira in their kidney. Is it fair to say, in the title, that IFN1 protects against renal colonization? Isn’t it more accurate to say that it reduces colonization of the kidney?

Methods:

Line 333: what test was done to make sure dead Lepto remained dead? Re-growth?

Reviewer #3: (No Response)

**Part III – Minor Issues: Editorial and Data Presentation Modifications**

Reviewer #1: (No Response)

Reviewer #2: 1.Author Summary: the first paragraph is the same as in Abstract, it could be deleted and the message in this section can be more straight forward.

2.The Introduction could be shortened to focus on how these studies advance the field. The first paragraph is redundant in the literature.

3. All figures: Captions need to be added to the figures for clarity. Regarding the clarity of the graphs, what does US mean in the axis X of graphs of Figure 1, 3, 4 and S1? Is it untreated samples? Is it possible to show the abbreviation in the legends?

Reviewer #3: The statistic analysis of Figure 1 data was not indicated. Does the expression of TNF, IL10 and IL5 significantly increase after 4, 6 and 8 h?

I believe this study was previously approved by the institutional ethics committee. Please mention that and the number of this certificate.

Please revise the format of references.

PLOS authors have the option to publish the peer review history of their article (what does this mean? ). If published, this will include your full peer review and any attached files.

**Do you want your identity to be public for this peer review?** For information about this choice, including consent withdrawal, please see our Privacy Policy .

Reviewer #1: No

Reviewer #2: No

Reviewer #3: No

**Figure resubmission:**

**Reproducibility:**



---

## [Decision Letter · Decision Letter 1]

29 Dec 2025

Dear Dr. Sutterwala,

We are pleased to inform you that your manuscript 'cGAS-STING dependent type I IFN reduces Leptospira interrogans renal colonization in mice.' has been provisionally accepted for publication in PLOS Pathogens.

Best regards,

Dana J. Philpott

Academic Editor

PLOS Pathogens

Thomas Guillard

Section Editor

PLOS Pathogens

Sumita Bhaduri-McIntosh

Editor-in-Chief

PLOS Pathogens

orcid.org/0000-0003-2946-9497

Michael Malim

Editor-in-Chief

PLOS Pathogens

orcid.org/0000-0002-7699-2064

Thank you for revising your manuscript and addressing most of the concerns of the Reviewers. Congratulations on your work.

Reviewer Comments (if any, and for reference):

Reviewer's Responses to Questions

**Part I - Summary**

Reviewer #2: (No Response)

**Part II – Major Issues: Key Experiments Required for Acceptance**

Reviewer #2: (No Response)

**Part III – Minor Issues: Editorial and Data Presentation Modifications**

Reviewer #2: (No Response)

PLOS authors have the option to publish the peer review history of their article (what does this mean? ). If published, this will include your full peer review and any attached files.

**Do you want your identity to be public for this peer review?** For information about this choice, including consent withdrawal, please see our Privacy Policy .

Reviewer #1: No

Reviewer #2: **Yes: ** Maria Gomes-Solecki

---

## [Editor Report · Acceptance letter]

Dear Dr. Sutterwala,

We are delighted to inform you that your manuscript, " 

cGAS-STING dependent type I IFN reduces Leptospira interrogans renal colonization in mice.," has been formally accepted for publication in PLOS Pathogens.

Best regards,

Sumita Bhaduri-McIntosh

Editor-in-Chief

PLOS Pathogens

orcid.org/0000-0003-2946-9497

Michael Malim

Editor-in-Chief

PLOS Pathogens

orcid.org/0000-0002-7699-2064